# Access to Landscape Finance for Small-Scale Producers and Local Communities: A Literature Review

**Bas Louman** [1,*], **Erica Di Girolami** [1], **Seth Shames** [2], **Luis Gomes Primo** [1], **Vincent Gitz** [3], **Sara J. Scherr** [2], **Alexandre Meybeck** [3] **and Michael Brady** [3]

1 Tropenbos International, Horaplantsoen 12, 6717 LT Ede, The Netherlands
2 Ecoagriculture Partners, Oakton, VA 22124, USA
3 Center for International Forestry Research (CIFOR), Situ Gede, Bogor Barat, Bogor 16115, Indonesia
* Correspondence: bas.louman@tropenbos.org

**Abstract:** Access to finance is a key element of sustainable and inclusive landscapes. We conducted a literature review to identify the factors that contribute to or hinder inclusive financing for micro/small/medium-sized enterprises and projects across sectors in ways that collectively contribute to more sustainable landscapes in the tropics. The key factors in the design of inclusive landscape finance are landscape governance, the financial literacy of local stakeholders, access to finance technology and services, and inclusive finance facilities and associated mechanisms for integrated (i.e., multi-project, multi-sector, spatially coordinated) landscape finance. The most frequent challenges are the types of existing financial products, the lack of livelihood assets among recipients (such as capital and income), the lack of transparency in finance mechanisms, the small scale of potential business cases, and the high risks perceived by finance providers and their customers. From this review, we propose components specifically focused on financial inclusion that complement the framework for integrated landscape finance developed by the Finance Solutions Design Team for the 1000 Landscapes for 1 Billion People Initiative. We suggest how the revised framework can be applied in designing and assessing the inclusiveness of finance mechanisms for integrated landscape management and to guide further research.

**Keywords:** sustainable finance; inclusive finance; resilient landscape; integrated landscape management; landscape finance assessment framework

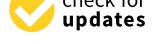



## 1. Introduction

The Finance Solutions Design Team for the 1000 Landscapes for 1 Billion People Initiative developed a framework for integrated landscape finance [1]. This paper proposes to enhance that framework with elements that will make integrated landscape finance more accessible to communities, small-scale producers and micro-, small- and medium-sized enterprises.

Using land effectively for crop production, grazing, timber extraction, and conservation is key to addressing humanity's basic needs. It is also central to solving the global crises involving biodiversity loss, threats to food security and nutrition, and climate change. The IPCC estimates that improving the management of such land use can abate up to 40% of total cumulative greenhouse gases by the end of this century [2]. Agriculture and forestry are also key to achieving most of the Sustainable Development Goals (SDGs) [3,4]. However, the finance gap entailed by the SDGs related to land use is hundreds of billions of USD per year [5,6].

Recently, large amounts of finance have been pledged for investments in the improved management of agriculture, forests, and wetlands. To leverage more private sector finance in investments that support the achievement of the SDGs, innovative finance structures and mechanisms have been designed, such as blended finance and green bonds. In order to

further increase the positive impact of their investments, an increasing number of investors—impact investors—include landscape considerations in their investment decisions. Where this is linked to strong local governance, it is leading to investments in a range of assets and sectors that complement each other in terms of positive impacts: integrated landscape finance [1].

Rural communities and small-scale producers play important roles in achieving the SDGs (for example, SDGs 2, 12, 13, and 15) and in addressing some of the global crises that threaten their livelihoods. They produce over 33% of the world's food [7] and manage 25% of the world's forests [8]. Investments in agriculture and forestry, therefore, need to consider the aspirations and needs of these small-scale producers and local communities in order to optimize the contribution of these sectors to the SDGs and climate goals. To date, however, most new finance pledges focus on meeting global goals and often require large-scale, acceptable risk-adjusted rates of return and complex forms of organization to be effective [9]. These, however, can rarely be applied to communities and small-scale producers in rural areas in the tropical South, who are more concerned with local needs and aspirations, which, therefore, are usually of smaller scale.

### 1.1. The Challenges of Financing Small-Scale Producers, Community-Based Enterprises, and MSMEs

Microfinance can play a crucial role in making finance accessible for individual households. However, despite the surge in microfinance during recent decades, it has barely contributed to the transformation to sustainable land-use practices (e.g., those that lower greenhouse gas emissions, maintain or enhance biodiversity and locally essential ecosystem services, or contribute to income and well-being). Many farmers who want to improve their agricultural practices require funding that goes beyond the financial instruments offered by microfinance institutions. At the same time, conventional finance institutions and even impact investors perceive the risks of such funding to be too high and the scale to be too small. This means that small-scale producers and communities that want to scale-up good production or conservation practices fall into what has been called the "missing middle" [10].

Balancing economic, social, and ecological interests and leaving no one behind requires the development of finance mechanisms that are locally based, meet local needs and aspirations as well as global objectives [11], and combine the benefits of global finance pledges with those of microfinance initiatives [12]. Only when also meeting local needs can the opportunities offered by innovative finance take full advantage of the potential contribution of small-scale producers and local communities in achieving the SDGs, the climate goals of the Paris Agreement.

To increase the positive social and environmental impacts of their investments, some agricultural and forest investment initiatives have applied a landscape approach. They consider the landscape as the scale where a range of land uses interact and, therefore, where the development and conservation impacts of their decisions can best be understood, managed, and assessed. Examples are the Tropical Landscape Finance Facility [9], the Tropical Asia Forest Fund [13], and the production, protection, and inclusion approach of the Sustainable Trade Initiative [14]. These and other such initiatives provide promising opportunities to integrate development and conservation objectives at the landscape scale [15], but, to date, they have been able to solve only some of the challenges of addressing local needs and aspirations, especially those of low-income groups [9].

### 1.2. Objectives and Organization of the Review

The objective of this review is to provide a critical analysis of scientific and non-scientific publications and gray literature on investments in agricultural and forest production and conservation within a tropical landscape context. It focuses particularly on lessons learned about the inclusiveness of initiatives and identifies those issues that need to be considered when designing inclusive financial mechanisms at the landscape level.

"Inclusiveness" here refers to facilitating access to finance (i.e., receiving credits for viable projects) for small-scale producers, community-based enterprises, and micro-, small-, and medium-sized enterprises (MSMEs, as defined by the International Finance Corporation) [16] that have a direct link with local agricultural or forest production. We use the term "MSMEs" to describe the entirety of this target group. Addressing the lack of inclusiveness needs to deal with the bottlenecks encountered by MSMEs (recipients) and by the providers of finance in the process of financing viable agricultural and forest investments.

The review provides information to assist NGOs, companies, financial institutions, international organizations, and government agencies—as well as landscape partnerships involving those actors—in designing instruments and products that finance resilient landscapes. This information can also support those farmers and communities who are not reached through large-scale innovative finance models, while addressing the impacts of the three global crises involving biodiversity loss, threats to food security and nutrition, and climate change. This is particularly relevant for initiatives that are designed to productively restore degraded lands in landscapes with many farmers and communities and with fragmented or unclear land rights.

Based on the literature review, the authors set out to answer the following questions:

- What are the challenges for inclusive integrated landscape finance as a path to resilient landscapes?
- What innovations are being used to address those challenges?
- What can be learned from successful innovations and how can these lessons be applied in the design and assessment of mechanisms for landscape finance?

This review starts in Section 2 by describing the focus area, integrated landscape finance, and a conceptual framework derived from the work of the Finance Solutions Design Team of the 1000 Landscapes for 1 Billion People (1000 L) Initiative [1]. Section 3 describes the review methodology, and Section 4 presents the results. Section 4.1 discusses the characteristics of the literature reviewed. Section 4.2 describes the challenges faced by MSMEs in achieving greater accessibility, differentiating the challenges according to major stakeholder groups (recipients and finance providers). Section 4.3 discusses examples of overcoming such challenges. This analysis leads to a discussion in Section 5 of lessons learned and suggestions for ways to enhance the 1000 Landscapes framework and increase its potential to be inclusive. The Conclusions (Section 6) summarize the main findings and their implications for further research.

## 2. Integrated Landscape Finance

We understand integrated landscape finance (ILF) as aiming to "support multi-project, multisector investment portfolios that encourage synergies between investments to generate impacts at scale across multiple landscape objectives" [15] (p. 2). The ILF approach can address sustainability challenges by building on and combining related initiatives, such as impact investing, conservation finance, blended finance, and inclusive green growth [15]. The approach is ever more relevant given the increasing financial commitments related to climate, landscape restoration, and biodiversity. In ILF, specific investments are designed and evaluated in the context of a broader landscape investment portfolio. This multisector portfolio is a set of activities that, if implemented in a coordinated way, generate value in and regenerate an ecologically degraded and economically unsustainable or impoverished landscape. The ILF portfolio is based on the idea that, by sequencing and spatially coordinating investments at a landscape scale, individual projects can be designed to increase synergies and reduce trade-offs. In this way, each project can achieve a higher rate of return, a lower risk profile, and/or increased social and environmental benefits. This approach can add value to individual investments by reducing risks, ensuring synergies and strengthening trust in project quality [15].

In 2021, the 1000 Landscape Finance Solutions Design Team developed a landscape finance framework [1] to guide the design and implementation of landscape transformation strategies at scale and to advance private, public, and civic investments and activities to

this end. To achieve such a transformation, five aspects must be addressed at the landscape scale (Figure 1):

- Developing a strategy to meet the long-term vision of the landscape partnership'
- Defining prospective projects that support the strategy (i.e., a set of assets and enabling investments that together can transform the whole landscape);
- Designing or incubating businesses/projects or scaling key landscape investments;
- Identifying or designing finance mechanisms to support the portfolio of investments and projects and components of them;
- Securing financial resources for the landscape investment portfolio that contribute to the transformation strategy.

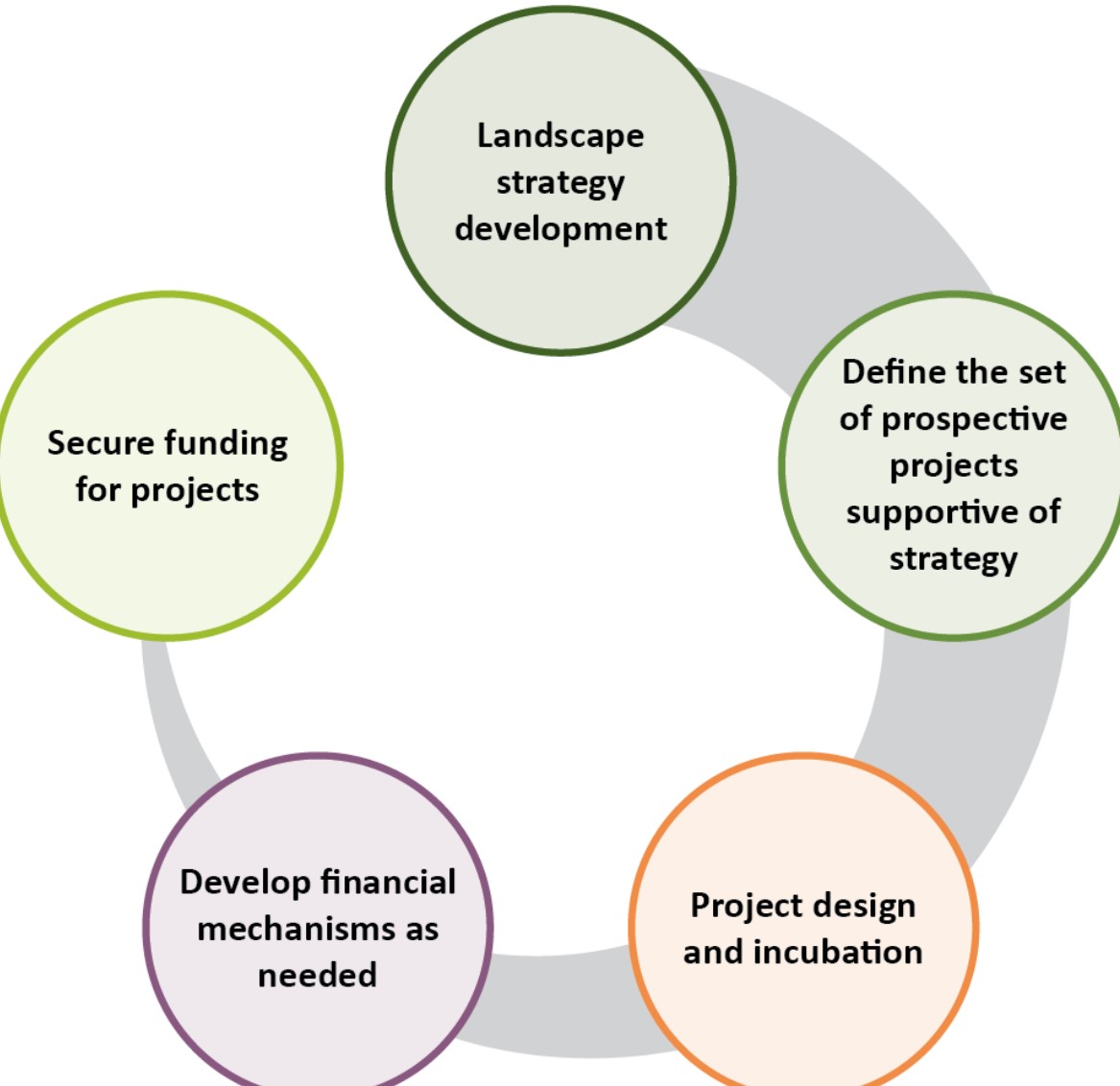

**Figure 1.** Framework for integrated landscape finance. Source: adapted from the 1000 Landscapes Finance Solutions Design Team [1].

Implementing such a framework first and foremost requires a strong multistakeholder landscape platform or partnership. Ideally, in such a partnership, civil society organizations (CSOs), local authorities and other relevant governmental entities, the private sector, and MSMEs collaborate to define a strategic vision for the landscape. This vision should identify public and private investment opportunities, including those enabling invest-

ments that are needed to attract investors. An entity or autonomous body with financial expertise then needs to structure appropriate instruments to provide these stakeholders with access to finance. These instruments need to be attractive to a range of relevant financiers and investors. This system needs to be accompanied by various risk reducing measures, such as guarantees, and must comprise an effective mix of grants and lending and equity instruments.

Integrated landscape finance is not just a strategy with a range of activities. It is also important to consider who designs and implements these activities and how. Shames and Scherr describe and analyze several models that have been used in landscape finance, most of which can facilitate finance for production and conservation [15]. However, most of these models were developed to address global or national goals, and they do not always address local needs and aspirations, partly because many originated from actors outside the landscape [9]. Most of the finance mechanisms developed to support sustainable landscapes are oriented around specific products and are not flexible enough to include those local stakeholders who seek alternative means of earning a living. This integrative literature review proposes that integrated landscape finance needs to become more inclusive.

## 3. Methodology

Distinct from other types of literature reviews, which focus on reviewing and summarizing specific scientific studies, an integrative literature review (ILR) is a research method that reviews, summarizes, and also critiques both empirical and scientific literature to provide an in-depth understanding of different aspects of a particular subject [17,18]. ILRs can draw lessons from both mature and emerging topics to generate new frameworks and perspectives [19].

For mature topics, an ILR allows for potential reconceptualization and expansion of a theoretical framework and can generate a new and significant understanding [19,20]. For emerging topics, an ILR creates a preliminary, theoretical understanding by collecting data and analyzing articles from multiple fields and research traditions [18,19].

An ILR requires a transparent and reproducible protocol to extract information and critically appraise the quality of the publications studied in order to gather high-quality data. Since finance for resilient landscapes is an emerging topic, the goal of this ILR was to contribute to the development of a theory—based on empirical and scientific evidence—of the factors that foster the success of finance mechanisms for resilient landscapes in the tropics.

We carried out the research in online databases from 28 June 2021 until 23 July 2021, mainly in Scopus for the scientific literature and Google for the gray literature. These were the inclusion criteria:

i.　Geographic focus. Priority was given to publications that focused on the Global South. Studies carried out in other landscapes could be included in the ILR if their findings were relevant to tropical landscapes;

ii.　Date of publication. Only studies published from 2010 onwards were considered. Exceptions could be made if the study was relevant for the ILR;

iii.　Language. Publications had to be in English, French, or Spanish;

iv.　Publication type. We included peer-reviewed journal articles, peer-reviewed book chapters, working/discussion papers, reports, policy briefs/ notes, and project notes.

The key search terms used to look for publications included "Sustainable Finance", "Innovative practices in finance", "Resilient Finance", "Inclusive Finance", "Integrated Landscape Finance Vehicles", "Value Chain Finance", "Conservation Finance Mechanisms", "Financial Gaps", "Inclusive Climate Finance", Climate finance for nature based solutions, Climate finance for community forestry, "Climate finance for tropical landscape initiatives", Climate Finance for Agroforestry, "Finance for Regenerative Landscapes", "Landscape Vulnerabilities", "Resilient Landscapes", "Inclusive Landscapes", "Integrated Landscape Management", "Climate-Smart Landscapes", "Landscape Initiatives", and "Regenerative

Landscapes". Some of the above terms were searched for without quotation marks because the combined word string did not generate any results.

This search identified 982 potentially relevant publications. We screened the abstracts for these publications, applying the same criteria and adding:

v.    Topic assessed and scale of analysis. We excluded all publications that focused on finance for renewable energies, transportation, cities, and infrastructure. Scientific papers concerning finance at the household or individual level were excluded as well, as we were interested in exploring finance at the landscape scale;

This led to 141 publications being selected for full-text reading. Several scientific papers—both open access and otherwise—could not be accessed because the publications were not available online. Eventually, 35 publications made it to the final sample for the actual review. In addition to the criteria noted above, we obtained this final sample by adding one specific criterion: "Does it address the challenges of financing integrated landscape management?"

After the search, and before reviewing the publications, a four-part data extraction form (DEF) was developed to facilitate information gathering:

i.    General study details (i.e., author(s), title, year of publication, academic journal, and database);

ii.    Study methodology (i.e., database, publication type, data collection type, sample size, data analysis, robustness of methodology, geographic focus, country, case study, general focus, specific focus on landscape, specific focus on finance);

iii.    Challenges for and factors in success (i.e., governance and institutions hampering positive outcomes, contextual factors hampering positive outcomes, any other challenges, trade-offs, specific financial gaps, governance and institutions fostering positive outcomes, contextual factors fostering positive outcomes, any other enabling factors, and synergies);

iv.    Quality assessment. The quality of the reporting was evaluated based on a quality assessment form inspired by Nyambe et al. [21] (Table 2. The assessment was based on the following indicators: clarity of research questions/hypothesis/study aim; clarity of data collection methods; clarity of sampling plan; clarity of sampling size; clarity of analysis method; clarity of conclusions; clarity of limitations; citations; and ability to cross-reference. Each quality indicator allowed for a score from 0 to 2, except ability to cross-reference, which allowed a score from 0 to 1.

After completing the literature review, we gained a greater insight into the topic and found that several aspects of the challenges, and the strategies that reduce them, needed further explanation. In response, we revisited references from an earlier report [9] and used additional articles and reports that were released after the review was completed. For the latter, we looked for cases that extensively described mechanisms seeking financial inclusion of MSMEs in the landscape, their challenges, strategies to overcome the challenges, and impacts of the financial flows that resulted from these mechanisms.

Figure 2 summarizes the literature review according to the sources used.

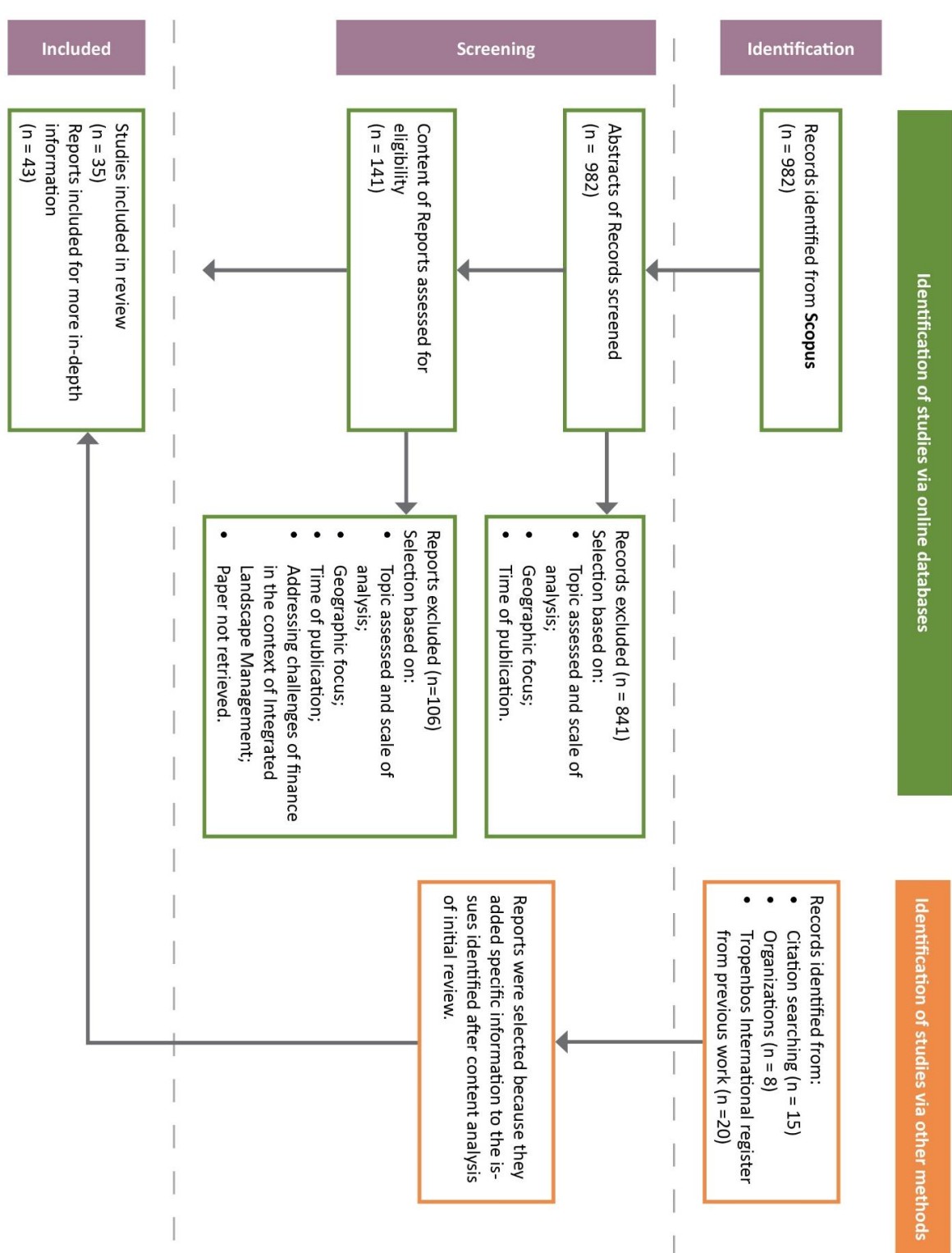

**Figure 2.** Overview of literature found in various sources for each search term and selection criteria used (modified from [22]).

## 4. Results and Discussion

### 4.1. Characteristics of Literature Reviewed

In the Scopus search, the search term "sustainable finance" generated the most responses (290, after eliminating double counts), followed by the terms "inclusive finance" (107), "financial gaps" (103), and "resilient landscapes" (102). Of the other terms, only

"integrated landscape management" (75), "inclusive climate finance" (53), "landscape initiatives" (35), and "inclusive landscapes" (24) generated more than 20 valid responses. Only 35 publications addressed the challenges to finance in the context of integrated landscape management and thus made it to the final sample. Of these, 28 publications focused primarily on finance, 3 focused on landscape, and 4 concerned landscape and finance. Most of the publications were published from 2018 on (n = 26).

The medium score for the sum of scientific quality indicators for the 28 publications that focused on finance was 13 out of a maximum of 17. For the publications on landscape, it was 7, and for the publications on landscape and finance, it was 11.

The geographic focus of the 35 publications was mainly tropical and cross-regional (n = 19), followed by Africa (n = 7), Latin America and the Caribbean (n = 5), and Asia and the Pacific (n = 4). The thematic focus of the publications on finance was diverse, including access to finance for cocoa production (n = 1), climate finance (n = 2), conservation finance mechanisms (n = 4), financial challenges (n = 1), inclusive finance (n = 10), innovative practices (n = 2), resilience finance (n = 1), sustainable finance (n = 2), and value chain finance (n = 3). The publications focusing on landscapes and relevant to the review explored themes such as innovative landscapes (n = 1), integrated landscape management (n = 1), and sustainable landscapes (n = 1). The publications on landscape and finance investigated subjects such as finance for integrated landscape management (n = 3) and for sustainable landscapes (n = 1).

### 4.2. Types of Challenges

Not all publications explicitly mentioned challenges for inclusive landscape finance. Some did so implicitly by proposing financial mechanisms to increase such finance; it was not always clear which specific challenges were being addressed by these mechanisms. The 26 publications that explicitly or implicitly mentioned challenges most often did so in relation to the characteristics of financial products and the prior access to livelihood assets of the intended recipients. In addition, the lack of mutual understanding between the financial sector and MSMEs was mentioned frequently (Table 1).

Several authors considered that short-term thinking, which tends to favor financial returns over social and environmental benefits, is a major challenge for inclusive finance for sustainable tropical landscapes [23,24]. Other challenges are local stakeholders' visions and priorities being undermined by top-down one-size-fits-all development strategies [25,26], existing policies and regulations [9,23,25,27,28], and tenure arrangements [29,30].

**Table 1.** Main challenges for landscape finance by category of stakeholders, as identified from the literature reviewed.

| Category of Stakeholders | Challenges | Authors | No. of Mentions |
|---|---|---|---|
| Recipients of finance | A. Product characteristics | [9,15,23,24,27,29–43] | 20 |
| | B. Livelihood assets | [9,15,23,27,29–40,44,45] | 18 |
| Finance providers | C. Knowledge of agriculture and forestry sectors and of landscape finance | [9,27,33,36,42,43] | 6 |
| | D. Scale, costs of services | [9,15,23,24,26,29–32,34,36,41,46] | 13 |
| | E. Transparency, trust, governance | [9,15,24,26–30,32–35,37,38,40,43,45,47] | 18 |
| Cross-cutting—both finance recipients and providers | F. Production, climate, price, and policy risks | [9,24,27,29,33–39,42,43,46,47] | 15 |
| | G. Lack of information, communication, roads | [9,23,27,29–32,34,39,41,42,44,45] | 14 |
| | H. Other benefits, policies, and regulations | [9,15,23,24,26–31,35,37,40,45–47] | 16 |

### 4.2.1. Main Challenges for Recipients

The main challenges identified for recipients were lack of financial literacy, insufficient technical know-how about transforming conventional land-use practices into sustainable practices, lack of collateral (often related to insecure land tenure), lack of access to financial institutions, insufficient capital or income, and poor organization. All of these challenges limit access to finance for MSMEs. We grouped these limitations together as limited livelihood assets, following Scoones' framework for the analysis of sustainable rural livelihoods, which distinguishes between human, economic/financial, social, and natural assets [48].

Insufficient technical know-how, lack of collateral or capital, and limited access to financial services have long been recognized as factors that limit access to finance for MSMEs and thus also limit the scaling-up of their activities. Financial literacy (a human asset) and support for financial literacy [29,31] have been recognized more recently as prerequisites to financial inclusion for rural and low-income people. Even basic skills—such as how to open a bank account, manage financial spreadsheets for keeping track of profits and losses, and use digital banking services—are often lacking in our target group [29]. Community business acumen, which has a strong positive impact on financial access and on the successful implementation of projects, is usually lacking as well [44]. Generally, people younger than 40, in particular women, need financial training [41,49,50]. Staff within many MSMEs need skills in tasks such as interest rate calculation, essential concepts of economic activities, and bookkeeping [23,29].

Training in financial literacy is often oriented towards understanding the use and implications of existing financial products. For many producers in tropical rural landscapes, however, these financial products do not meet their needs or circumstances. Examples of mismatches include [23,24,27,29–35,37–39,41–44]: lack of collateral due to insecure land tenure; payment periods that do not match harvesting cycles; loan terms that do not match production cycles; and administrative processes that are not transparent or predictable, which leads to people not receiving loans when they are most needed. These mismatches are partly due to a lack of knowledge about the agriculture and forestry sectors and their needs among financial institutions and regulators.

### 4.2.2. Main Challenges for the Finance Providers

The major challenges for finance providers (FPs) include an incomplete understanding of the agriculture and forestry sectors, making it difficult for them to build relations of trust with the various stakeholders [27,33,43]; the limited size and number of familiar and proven business cases; and, as a result, the relatively high costs of providing services [23,29,30,32,45]. Another challenge is the FPs generally poor understanding of what it takes to make finance more inclusive of MSMEs. Challenges to inclusiveness include regulations that limit the flexibility of finance providers in offering tailor-made solutions to local stakeholders [23,27] or that limit the capacity of MSMEs in the agriculture and forestry sectors in fully participating in national and international markets [25]. Other financial infrastructure can also be limiting, such as lack of credit and collateral registries. Being able to provide locally appropriate financial instruments appears to be particularly relevant in the forest sector, both for small-scale tree planting [51] and for managing natural forests [52]. These challenges make it difficult to identify robust and investible projects within the landscape. This is often worsened by cross-cutting challenges (Table 1 and Section 4.2.3); in particular, the FP's perception of high risks in agriculture and forest-sector investment [9,15,27,32].

### 4.2.3. Cross-Cutting Challenges

FPs and recipients all perceive some level of risk, although they differ in the type and level of the risks perceived. Many recipients are producers and often perceive production risks related to weather, pests and diseases, or their own health [53]. Although production risks may affect all stakeholders—because the recipients would not be able to pay back their loans if production were compromised—stakeholders are affected to varying degrees. For

example, FPs may be able to reduce their risks through insurance schemes or guarantees. Crop insurance for producers is less common, however, and is rarely accessible to MSMEs in tropical countries [53,54]. In addition, large agribusinesses or international FPs can invest in production across a range of geographical areas or products, thus lessening their risks.

FPs are more focused on financial risks (such as inflation, changing conditions for accessing credit), market risks (such as trade challenges, fluctuating exchange rates), client risks (creditworthiness, integrity, due diligence), and national and international policy risks (such as environmental, monetary, and trade policies, which may affect the value of ecosystem services, such as carbon or water; and the need to meet social and environmental safeguards) [35,41,42,53,55,56].

Recipients also pay attention to market risks (price fluctuations) but less attention to financial risks (fluctuations in income or access to finance) or risks related to institutional or policy changes. Linking recipients to secure, well-paying markets for a new crop or product (and thus reducing their market risks) may convince them to accept the production risks and make the shift. This happened, for example, with oil palm in Indonesia; it is replacing rubber as a commercial crop because of its higher and more stable prices [57].

Lack of access to information is another barrier reported for both recipients and providers (see Table 1). This affects recipients in relation to weather conditions, technology, good practices [34,39], markets, access to FPs [44,45], and information sharing by stakeholders in the same value chain [35]. This challenge is particularly important in landscapes that are subject to rapid changes from, for example, climate change. MSMEs are more affected by lack of information than large companies, who usually have more resources to overcome this barrier [58]. This challenge is exacerbated by inadequate communication infrastructure [32,41].

Poor quality roads [30] and other types of infrastructure [59] also limit MSMEs in scaling up their activities and, therefore, impose obstacles for inclusive finance.

### 4.3. Overcoming the Challenges in Accessing Integrated Landscape Finance

A range of factors can help overcome the challenges of inclusive landscape finance. They can be grouped in four broad domains (see Figure 3):

i.      Inclusive landscape governance (addressing trust, transparency, and regulations) [9,15,24,26,27,30,40,47,60] and collaboration between local stakeholders [9,29,32,35];

ii.     Strengthened financial literacy of MSMEs (addressing financial asset creation, business case development, and marketing) [9,29,40,44,49];

iii.    Access to finance technology and services (addressing the needs of MSMEs for access to FPs, market information, technical assistance, and technological innovations) [31,34,35,39,41,44,45,61];

iv.     Inclusive facilities and mechanisms for integrated landscape finance (addressing the costs of financial services [30,32,50,61], product characteristics [9,31,32,34,35,38,39], incorporating a greater range of benefits [15,27,29,30,42], and risk management [9,15,36,39]).

### 4.3.1. Inclusive Landscape Governance and Local Stakeholder Collaboration

Landscape governance requires institutions to trigger systemic change by including all the relevant stakeholders—from the private sector, public sector, and civil society—throughout the landscape [9,15,26,27,47,60,62]. Inclusive landscape governance helps clarify rules and regulations and facilitates their implementation.

Inclusive landscape governance has been instrumental in developing alternative forms of collateral, based on mutual trust and social control rather than on formal property rights [57]. In addition, formalization of property rights stimulates investment and decreases the risk that communities will lose access to their lands to more powerful actors [24,30,40].

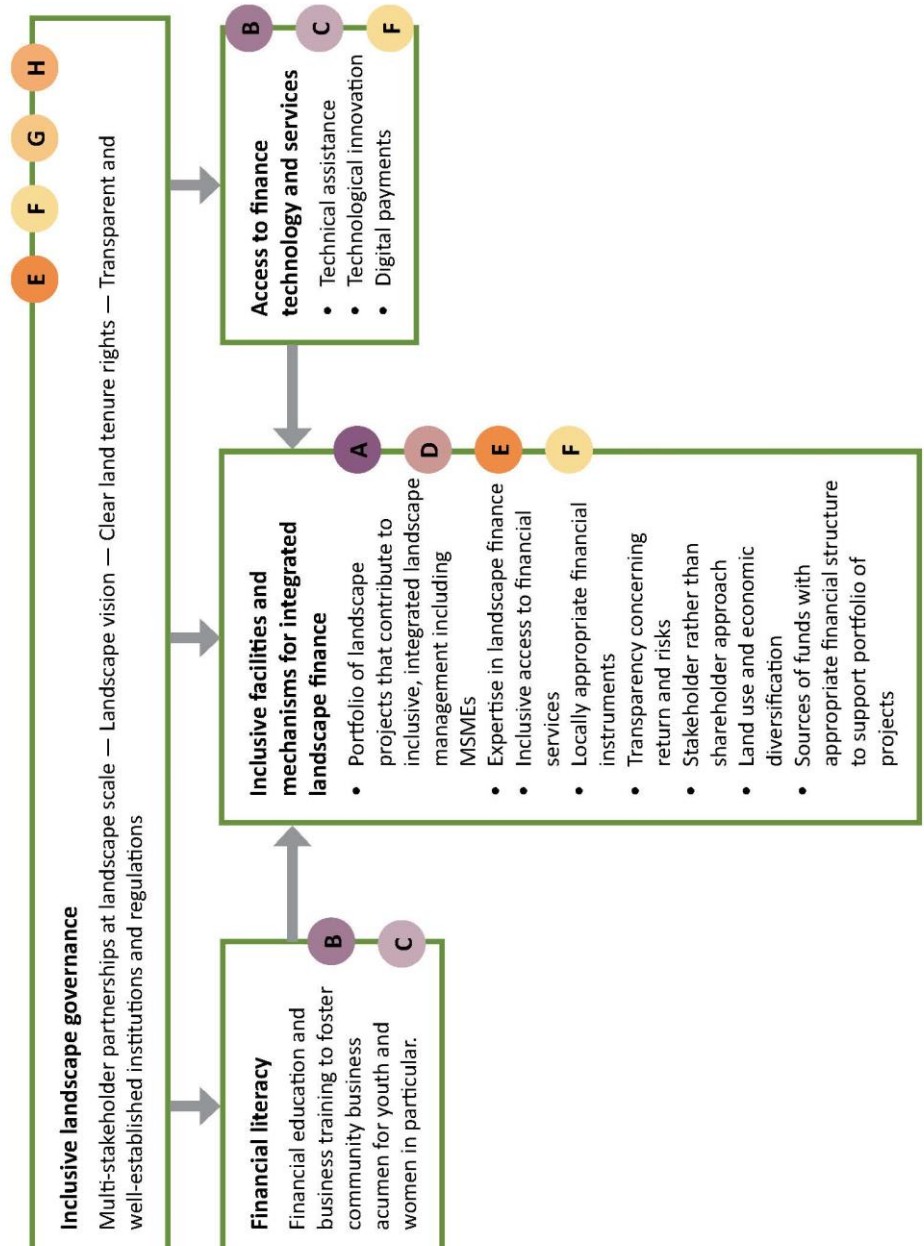

**Figure 3.** The four domains that facilitate the inclusion of MSMEs in landscape finance. Legend: The circled letters represent the challenges listed in Table 1 that are addressed by each domain: **A** product characteristics; **B** livelihood assets; **C** knowledge of agriculture and forestry sectors; **D** scale and costs of services; **E** transparency, trust, governance, and financial infrastructure; **F** production, climate, price, and policy risks; **G** information, communication, and roads; **H** other benefits, policies, and regulations.

Inclusive governance also supports collaboration among stakeholders with common objectives. Multi-stakeholder approaches are widely recognized as essential for integrated landscape management [63,64] and for financing ILM [24,27,30], particularly to achieve synergies and balance trade-offs between investments within the landscape. The literature highlights the need for strong partnerships with value chain actors and well-established groups who understand local conditions and are part of decision-making processes [30,32,45]. Decision making also needs to be transparent about who benefits and how in order to avoid discontent among recipients [65].

For example, partnerships among a range of actors can enable coordinated action and benefit-sharing agreements in restoration schemes funded by carbon finance, contributing to their transparency and effectiveness [51,66].

Collaboration is necessary to achieve the larger scale of investment and impact that most FPs and investors are looking for. It can help create diverse portfolios of investable projects [15,27] or group MSMEs in forest and farm producer organizations in order to produce, store, process, and/or sell their products [9,29].

Despite the importance of landscape governance and multistakeholder approaches to integrated landscape management and its finance, none of the studies reviewed indicated that financial decisions were guided by an effective and efficient multistakeholder platform. The mere existence of a multistakeholder platform, therefore, is not sufficient. There needs to be a facilitating agent willing to and capable of involving all platform members in planning, financing, and implementing inclusive integrated landscape management. Many existing multistakeholder platforms fall behind in implementing their vision [67] due to limited-capacity or under-financed facilitation.

Current financial practices do not always align with perceived local priorities [66,67]. Often, investments are aligned with the objectives of the generating cash flows (from, for example, cocoa, rubber, timber, or carbon credits), but studies that went beyond the value chain analysis and also looked at landscape needs and aspirations revealed that investments often met only some of the landscape actors' other objectives [65,68]. Where a disconnect exists between local stakeholders and financial institutions, concerns may arise regarding lack of transparency or unfair benefit-sharing mechanisms [66] or because access to funds and technical support was available for only a limited number of farmers or a limited number of activities [68].

One study reported that inclusiveness in an agro-commodity value chain was greatly improved due to partnerships with international companies, but that economic diversification remained under-addressed; this particularly affected female members of farm households [68].

In addition to the governance of landscapes and value chains, the inclusiveness of landscape finance also depends on governance factors outside the landscape. This includes any agreements between local, national, and international actors and how these agreements are structured and being implemented [25,32]. For example, transparent and well-established public institutions improve the financial infrastructure and can decrease the challenges for small firms in obtaining finance [25]. Improvements of such financial infrastructure may include setting up the systems that enable mobile payments by strengthening creditor protection [69] or setting up transparent registries for movable collateral [70]. In some countries, this has enabled the use of trees as collateral [71]. Secure land tenure is often mentioned as a prerequisite for sustainable land use [72] and encouraging investments [30]. While local mechanisms may be designed that incorporate traditional governance and compensate for a lack of formal tenure [57], they are still rare, and access to finance remains limited to those who can comply with national regulations.

### 4.3.2. Strengthening the Financial Literacy of MSMEs

Financial literacy is not just a concern at the household level. The financial literacy of decision makers improves the performance of MSMEs [73,74], increases willingness to save and often leads to greater access to FPs [49,75].

Strategies to strengthen financial literacy may include setting up village savings and loans associations (VSLAs) [68] or working with microfinance institutions (MFIs) [23,29,31] and credit unions [57]. Many households and microenterprises have benefitted from membership in VSLAs [76,77]; in particular, when these associations were linked to financial training [78,79]. However, unless VSLAs are used as an entry point for credit unions, they may not provide sufficient funds to scale up finance for sustainable land use and related SMEs [38].

Training in financial literacy encompasses more than learning how to save, spend, and earn. The Organization for Economic Cooperation and Development and the International Network on Financial Education have developed a framework of core competencies that MSMEs need in order to be considered financially literate [80]: making the right choice of which financial services to use; financial and business management and planning; understanding risk and risk-reduction strategies; and understanding the financial context. The degree to which each of these competencies needs to be mastered will change as the organization or business evolves [81]. A VSLA may play an important role in the initial stages of such training, but to scale up sustainable economic activities its members usually will need a higher level of financial literacy. Strengthening financial literacy of SME managers, for example, can be addressed by providing capacity building in business skills [29].

One of the big challenges for local economic organizations is the internal governance of their finance, in terms of organizing production and sales (see Section 4.3.3) and the creation of fair benefit-sharing mechanisms. Such mechanisms may involve agreements on who works how often and for what salary [52] and may determine the percentage of income that goes to local community works or funds [82] or how income is distributed among the members of an organization or community. When defining the best benefit-sharing mechanism for an organization, it is important that the decision makers have a good understanding of the implications of the proposed mechanism for the financial health of the organization.

Women and youth rarely receive the special attention they require in financial literacy training [41,50]. This was confirmed in a study of a women's bond and crowdfunding platform in Asia [83]. This gap may be due to the lack of educational opportunities for young women and to women being less available for training due to their roles within the household [68]. To achieve inclusive landscape finance, financial literacy training will need to address the challenges for women and youth, specifically their opportunities to earn and save.

### 4.3.3. Access to Finance Technology and Services

Financial inclusion is often improved through FPs increasing the range of products and services they offer and by their being more responsive to the needs and aspirations of local people [31]. Using finance technological innovation (for example, blockchain technology) and digital payments appear promising for promoting financial inclusivity, mitigating risks, and helping communities in a cost-effective manner [9,31,35,84]. The feasibility of their application needs to be studied for each initiative [35], however. In many tropical landscapes, their widespread application in the agricultural sector requires additional technological improvements [85], as well as greater internet connectivity for potential users [61], making finance technology still of limited applicability for most rural MSMEs.

The use of digital technology to increase access to finance is not yet widespread, and physical access to services remains a constraint. In one case in Indonesia, a credit union compensates for its lack of a physical presence by employing rural representatives [57]. In Ghana, VSLA treasurers had to travel long distances to deposit the savings into a bank account; a major risk of assaults on the money bearers was perceived [68]. Under such conditions, incorporating digital technology in the design of local financial systems could facilitate the participation of formal financial institutions and thus help increase access to such institutions by rural people. In Indonesia, for example, the uptake of digital payments has increased since the introduction of a digital system (QRIS) in 2019 [86]. In sub-Saharan Africa, mobile money increasingly supports basic services and has contributed to the socioeconomic well-being of many people [87]. This is particularly well-developed in Kenya through the popular M-PESA digital banking system, but while it does benefit rural areas, its positive effects are greater in cities [88]. This indicates that the system needs further development to also serve the needs currently met by VSLAs and local branches of credit unions and banks.

Scaling-up access to finance is not just a matter of making finance available for investments. The importance of technical assistance for the development of investable initiatives should not be underestimated [9,35,83]. It is particularly relevant in the context of climate finance and international sustainability standards. To qualify for such finance, recipients need to make adjustments in existing or conventional production and conservation practices. Some authors recommend increasing finance for technical assistance to reduce both the technical and political challenges to financing this transformation [7].

Indeed, the increase in various forms blending development and commercial money has led to an increase in mechanisms that combine financial support with technical assistance. Commercial investments in large-scale agro-commodity enterprises are often combined with development money to finance technical assistance so that MSMEs can provide a higher quality and quantity of raw material. A good example is the Tropical Landscape Finance Facility (TLFF), which blends money from various types of investors with development money oriented towards improving rural livelihoods. This allows TLFF to also provide technical assistance and inputs to MSMEs related to the agribusiness (rubber) that the facility has invested in. Although this facility is still young, in 2021 it reported that approximately 1000 small-scale producers had participated in the program and benefited through an income increase of 30–50% since their participation [89]. Participation is limited, however, to rubber producers and communities that conserve forests; i.e., those that fit within the objectives of the agribusiness and its investors.

In some cases, small-scale producers and communities have organized themselves in associations, cooperatives, or credit unions to create a fund that provides access to both finance and technical assistance, independent of any link to specific product markets [9,57]. In other cases, investments are related to reduced carbon emissions, and the proceeds fund technical assistance to MSMEs to help them improve their agricultural production, reduce their need to expand into natural forests [90], and restore degraded areas [51].

In the context of these innovative finance structures and recent international sustainability demands, technical assistance and training should include diversification options. This should reduce the risk that farmers focus on one crop or product and therefore become more vulnerable to price and yield variations [91]. Due to the variety of training needs that diversification involves, it will usually require collaboration between local CSOs, extension agencies, and companies that provide training and technical assistance.

Intermediary organizations (e.g., public institutions, non-governmental organizations) were important providers of services in many of the studies that we reviewed. They provide technical assistance, information on production innovations, business training, and knowledge related to markets and other services. They may also provide mentoring programs, training for designing and developing bankable projects, and business incubation services and can foster networking services to link MSMEs with potential investors [29]. Financial intermediary organizations may also be able to help set up the most appropriate financial structures and products (see also Section 4.3.4).

In many of the cases we reviewed, the intermediary organizations that provided technical assistance collaborated with organizations responsible for integrated landscape finance systems. Together, they helped structure financial instruments that were appropriate to attract a mix of funders and to provide access to finance for technical assistance and productive or conservation projects.

### 4.3.4. Facilities and Mechanisms for Inclusive Integrated Landscape Finance

Inclusive landscape governance, appropriate levels of financial literacy, and access to technical and business innovations, markets, and services should link to—and catalyze—the development of finance facilities and mechanisms for inclusive integrated landscape investments.

**Landscape finance facility.** Key to any such facility is that it responds to the local needs of various rural sectors and population groups. It should bring together the demand for finance from landscape actors and community institutions with the supply of financial products by finance institutions. The review found descriptions of various financial mech-

anisms that allowed more finance to reach tropical landscapes, but challenges remain in making these mechanisms broaden access to more stakeholder groups and to innovations in agricultural and forestry production.

The improvements most often mentioned in the literature in this regard concerned an enabling institutional environment, technical assistance, and bringing together a variety of funding sources into specific financial instruments [9,15,51,92]. The latter were usually managed by a fund manager or project coordinator and used a range of strategies to address scale, risk, and investors' expectations of returns. Fund managers offered various types of access to funds, reaching the recipients through locally adapted financial products or by financing technical assistance or specific inputs. This implies that, in these cases, the FPs have knowledge of landscape finance, of economically vulnerable customers' needs, and of landscape-friendly business models.

**Investment diversity.** Despite these improvements, there is still a lack of diversity in the range of projects included, and many stakeholders face a lack of access to funds. Although the Tropical Landscape Finance Facility and the Tropical Asia Forest Fund (https://newforests.com.au/tag/taff/, accessed on 20 January 2022) provide important private finance for sustainable agriculture and forest operations in tropical landscapes, they are directed at making a major agro-commodity production more sustainable rather than supporting a broad and diverse portfolio of sustainable economic activities that address the needs of the people in those landscapes. Such initiatives are important steps in achieving sustainable landscapes and have great potential to contribute to global goals. However, they run the risk of excluding many MSMEs in the landscape that, for various reasons, cannot or do not want to get involved in the monocrop supply chain.

The review found several cases in which initiatives were tailored to the local context [66,68,82], but the financial or benefit-sharing mechanism focused on a single product and market, such as cocoa, wood, and carbon value chains. This made the producers more vulnerable to fluctuations in market access and price [66,68,82]. Female farmers or female members of households may be particularly affected by a focus on a single crop or product [68]. In a few cases, the implementing agency builds-in risk-reducing mechanisms (such as a buffer fund) or seeks diversification of land use and local financial instruments [51]. Investing in diversification of land use is in general a recommendation for biocarbon projects [30] but occurs less in initiatives related to agro-commodity value chains.

A facility for integrated landscape finance could be based on aggregating diverse demands for finance. It could support various bankable projects and provide a range of locally appropriate financial products in order to create a portfolio of sufficient size and diversity to attract a variety of investors [24]. This would achieve scale and diversifying the portfolio of activities to be financed would also reduce risk. Risk can be further reduced by various forms of co-investments, such as subsidies, guarantees, junior debts, or co-investments with landscape partners [29,92].

Creating an economically viable portfolio of diverse projects can be facilitated by supporting local actors in developing their business plans, thus creating a robust portfolio of project applications. The finance facility should also provide an effective coordination mechanism to ensure that the portfolio of projects aligns with landscape objectives [15]. To improve access to finance, project proponents need to learn how to meet financial institutions' requirements, and the financial institutions need to develop locally appropriate financial instruments [9]. The latter has been successful in cases where financial institutions or credit unions were locally present and, therefore, better understood the financial needs of the recipients [57]. Adjustments made by lending organizations to the type and requirements of their financial services include allowing contract agreements or buyer's guarantees to be used as collateral [32], allowing for group lending agreements [32], and providing loans for specific purposes, such as health and education, that are tied to existing savings [57,82].

**Market access**. Most MSMEs need to increase their income in order to be able to pay back any loan. In some cases, farmers may prioritize access to markets rather than

access to finance [57]. Value chain finance seems to be well-placed to address access to markets; several experiences show that, in practice, access to markets, finance, and technical assistance are closely interlinked [35,68]. Access to markets for a diversity of products may be one of the major challenges for inclusive finance for sustainable landscapes, particularly in less developed countries. Local markets are very limited because local income and production are low, while agro-commodity value chain companies usually focus on one single product and the farmers that provide it.

Increasing the inclusiveness of finance in such conditions requires investing in technical assistance that increases access to local and national markets; for example, by supporting start-up MSMEs to develop primary processing practices and facilities. This is an aspect that still receives relatively little attention in the reviewed literature, possibly because many studies involve linking smallholders to existing value chains in the landscape (agro-commodities, conservation of biodiversity, carbon).

**Innovative financial instruments.** Blended finance and green, social, climate, and sustainability bonds are evolving financial instruments. They aim to facilitate risk sharing and attract private investments to the transformation to sustainable, climate-resilient, and low-carbon economies. Blended finance structures, for example, can contribute to making value chains and trade more sustainable [36,93]. Combining development finance with commercial finance can support several enabling processes, such as strengthening landscape partnerships, building portfolios of projects, and improving the institutional context and access to markets. Such financing structures, therefore, seem to fit well with the concept of an integrated landscape finance mechanism [15]. Promising evidence exists of successful application in the agriculture and forestry sectors [13], and the Shames and Scherr survey [15] found 26 integrated landscape finance mechanisms; i.e., multi-project, multi-sector, coordinated across the landscape. Models being implemented or under design included landscape funds, multi-landscape funds, landscape bonds, and landscape development finance institutions, such as landscape banks. The use of such structures in the sector, however, lags behind that in other sectors and, in most cases, has only partly been able to address the issues of diversity and inclusivity [9]. The Livelihoods Fund for Family Farming (L3F, https://livelihoods.eu/l3f/, accessed on 21 July 2022) and Root Capital (https://rootcapital.org/, accessed on 21 July 2022) are exceptions that directly serve small-scale producers and/or processing MSMEs. The former applies a landscape approach to its investments and promotes aggregation at the investee level, while the latter works through local private businesses as aggregators and applies off-taker commitments as guarantees for loan repayments. In both funding structures, returns are directly related to results-based payments for products and environmental or social benefits, which are previously defined [13,15].

Many production risks can be abated through improved technology and technical knowledge. Some could also be covered by agricultural insurance. In most developing countries, however, the agricultural insurance market is not well-developed and many farmers seek alternative means of risk sharing (with relatives, for example) or risk reduction (through diversification) [54]. Despite advances in risk reduction mechanisms by both investors and recipients, the level of risk perceived by both sides remains a major challenge to inclusive landscape finance. It is likely that this perception could be changed by sharing more information on successful landscape investments and their risk reduction mechanisms.

**Stakeholder approach.** Several authors recommend that investors and financiers engage with multi-stakeholder platforms to build trust and address power imbalances [24,29,33–35,43,45] and to ensure that projects are relevant to the context and embrace the socio-cultural dimensions of the places where they operate [34]. When projects and business investments do not sufficiently integrate context, they can have significant negative impacts in the landscape where they are implemented [45]. Part of the responsibility of a facility would therefore be to promote such engagement and alignment with local stakeholders' landscape goals and action plans, not just with the objectives of the value chain or global imperatives.

A facility for inclusive integrated landscape finance should also consider a stakeholder—rather than shareholder—approach [27,46]; embedding the finance mechanism in a landscape governance structure [24,26]; recognizing the importance of inclusiveness of women and youth [29,30,34,39,41,83]; and monitoring and recording financial, social, and environmental impacts in a transparent way to make the portfolio of projects more attractive to investors [27,29,83].

By applying a stakeholder approach, implementing agencies can address the needs of the stakeholders and priorities in the landscapes rather than just the priorities of the shareholders. This allows for a longer-term perspective, which is especially important when dealing with forests, tree crops and restoration of natural infrastructure for biodiversity, watershed management, and climate change mitigation and adaptation [24,27,30]. It also allows implementers to design innovative financial products and services that match the needs and values of local stakeholders. In Indonesia, for example, a credit union designed its products in a way that contributed to education, self-reliance, solidarity, innovation, and unity in diversity, values defined by its members [57]. Thus, the credit union provided additional services, such as insurance for health, death, and house fires. Other studies highlighted the opportunity to integrate financial returns with other objectives, such as increasing access to finance for health and education [23], an important motivation for producers to participate in community-based funds such as VSLAs [68,76], community rotational funds [82], and credit cooperatives and unions [57].

## 5. Towards Inclusive Finance for Integrated Landscape Management

The review aimed to identify factors that contribute to the inclusiveness of existing innovative financial structures and mechanisms to create sustainable and resilient landscapes. Inclusive landscape governance, including strong multi-stakeholder partnerships, financial literacy, access to finance technology and services, and facilities and mechanisms for inclusive landscape finance all appear to be essential.

Together, these enable development and financing of a portfolio of bankable projects encompassing key aspects of a sustainable landscape, with full participation of smallholder farmers, community organizations, and MSMEs. It is important to create or strengthen the enabling conditions for such a system.

### 5.1. Institutional Development for Inclusive Landscape Finance

The four domains shown in Figure 3 were addressed in most of the cases we reviewed, independently of the source and structure of the finance. However, the main objective of inclusive finance for integrated landscape management—attracting multiple finance flows to a balanced portfolio of multiple projects that are aligned with local and national objectives—is not yet being achieved. From the literature and cases reviewed, two main approaches to inclusive landscape finance emerge: (i) top-down, from the outside into the landscape, addressing international or national sustainable development and environment objectives and/or value chains' objectives seeking to make local production sources more sustainable; and (ii) bottom-up, from the inside out, working mainly with local savings and non-commercial resources and directly addressing the needs of local stakeholders.

Currently, most finance initiatives focus on addressing the needs of investors and the international community (e.g., climate mitigation through carbon sequestration and avoided deforestation or conservation of biodiversity by using blended finance or climate bonds) rather than on meeting the needs of local people. Inclusive integrated landscape management must address both needs. In some cases, the first steps towards financing integrated landscape management have been taken [51,57,68]. However, in general, financing initiatives rarely include financial literacy training or technical assistance to support sustainable and diversified production, as these go beyond the capacities of the implementing agencies in terms of staffing, knowledge, and experience. New initiatives are being proposed to address this gap, such as the Green Finance for Sustainable Landscapes program (https://www.unep.org/fr/node/28994, accessed on 21 July 2022).

To achieve results at scale requires collaboration among a range of actors, both within and outside the landscape. One of the greatest challenges in the design and implementation of facilities for inclusive and integrated landscape finance is creating a network of actors who are willing and able to collaborate. Central to such a network are the local MSMEs, farmers, communities, and their farm and forest producer organizations—ideally connected through a multistakeholder platform or partnerships— as well as other participants:

- **CSOs** that build capacities in financial literacy, business acumen, and technical issues related to production and market access;
- **Local financial institutions** that develop locally appropriate financial instruments;
- **Fund managers** who translate the needs of the MSMEs into investible projects and design financial instruments through which public and private investors can invest in them;
- **Landscape finance support service providers** who can help connect MSMEs with FPs, coordinate and aggregate projects for synergies to meet landscape objectives, and incubate inclusive green businesses;
- In some cases, **knowledge platforms** through which fund managers can access investors and work with organizations that can provide the technical knowledge needed by local CSOs.

A close relationship between local stakeholders and financial organizations has been reported in the literature as one of the factors that leads to success. For credit unions in Indonesia, it was reported to be a factor that motivates the clients to comply with their commitments to "their" union [57,94]. Credit unions, however, are not necessarily linked to landscape objectives; in Indonesia, a credit union may finance initiatives that do not contribute to a landscape that balances production and conservation [57]. Including sustainability criteria in their loan application process could change that, but this effort may need external support and additional capacities for monitoring. In addition, credit unions do not have the funds necessary to upscale investments at the landscape level or to fund larger projects or investments in broader initiatives, such as biodiversity conservation.

Inclusive finance for sustainable landscapes requires a combination of locally led and externally driven approaches. An example is the Trees for Global Benefit program, implemented by Ecotrust in Uganda [51]. The program obtains carbon finance for planting and maintaining trees, thus addressing the objectives of the finance providers, who are outside the landscape. At the same time, it uses the funding as collateral to obtain loans for other activities that diversify the local economy in order to also meet the objectives of the recipients in the landscape.

Creating initiatives that in the mid- to long-term will attract private investments to support production, well-being, and conservation in the landscape requires time (for organization) and money (mainly for capacity building and technical support). Such efforts can be achieved only by blending public and private finance. This may require a revision of the current blended finance practices. Rather than combining development money with commercial investments, it requires that such money precede commercial investments and be independent of them to prevent development money focusing on a single crop.

*5.2. Enhancing the Landscape Finance Framework to Explicitly Address Inclusion*

We argue that, although the framework representing current integrated landscape finance (Figure 1) is useful for the domains of a landscape facility, it does not address the specific challenges of inclusions B (livelihood assets), C (knowledge of agriculture and forestry sectors and of landscape finance), G (lack of information, communications, roads), and H (other benefits, policies, and regulations) in Figure 3 and only partially addresses challenges A (product characteristics), D (scale and costs of services), and E (transparency, trust, and governance) in Figure 3. The results of the review point to the need to revise the framework in several ways. We suggest that, to increase inclusivity, a parallel inclusion pathway addressing specific inclusivity challenges should be added (Figure 4). The parallel pathway could include financial literacy and access to finance technology and services as

steps between developing, defining the set of prospective projects, supporting the landscape, and development of financial mechanisms, with continuous feedback between the two pathways. In addition, the framework should explicitly include inclusive landscape governance and inclusive landscape finance facilitation as overarching enabling conditions (see Figure 4). Inclusive governance is essential to ensure that more local stakeholders are able to contribute to the development of the landscape strategy and to develop bankable projects that will form part of an integrated investment portfolio. Inclusive finance facilitation is needed to coordinate and connect investment and finance efforts, including for smallholders, community organizations, and SMSEs. These modifications will enhance integration of the portfolio in terms of covering a range of economic activities, as well as a range of project sizes and contributions to various landscape objectives.

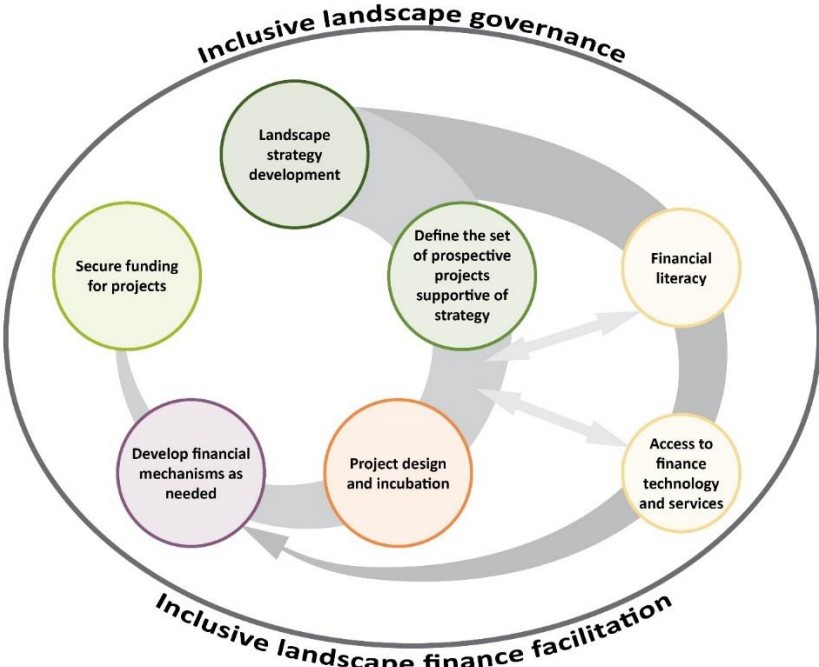

**Figure 4.** Proposed framework for inclusive integrated landscape finance (adjusted from Figure 1).

By including financial literacy and access to finance technology and services, the framework in Figure 4 explicitly supports the development of projects by local people. Financial facilities and mechanisms need to be developed that enable these projects to secure funding. This framework can be complemented by others that focus more on the development of specific financial mechanisms and on securing funding, such as the ones supporting community conservation projects reviewed by Smith et al. [92] or those applied in some of the blended finance models reported in the literature [13,15].

### 5.3. Implementing the Framework

Inclusive integrated landscape finance aims to align investments with positive social and ecological impacts, as well as economic gains, at both the project/enterprise and landscape scales. The proposed framework shown in Figure 4 can help policy makers, international donors, and academics evaluate whether the finance structure in a landscape includes the key elements that are necessary for the successful inclusion of smallholder producers, local communities, and their producer organizations.

Table 2 below lists a set of guiding questions to assess the inclusivity of the landscape finance mechanisms in place, derived from the literature review.

**Table 2.** Questions to assess the inclusiveness of landscape finance.

| Framework Components | Guiding Questions |
|---|---|
| Inclusive landscape governance | • Are all the relevant actors from the private sector, public sector, NGOs, and academia involved in a multistakeholder platform (MSP)?<br>• Do the members of the MSP have the expertise needed to address the various components of the framework (financial, social, technical)?<br>• Are power relationships within and among stakeholder groups managed to enable clear, transparent exchanges and open dialogue?<br>• Which groups are likely to be at a disadvantage in discussions?<br>• How are gender, cultural, religious and ethnic differences expressed? Are they recognized, respected, and considered in dialogue and in rights, and access to resources and reflected in cost- and benefit-sharing mechanisms?<br>• What are the local access and tenure issues?<br>• Are there mechanisms in place that allow for conflict management?<br>• What are the visions of the different stakeholder groups?<br>• Do stakeholders agree on the elements of a joint landscape vision?<br>• Is an effective regulatory framework in place?<br>• Are any well-established institutions operating in the landscape; for example, for monitoring and control of land use regulations?<br>• How transparent are these institutions for local stakeholders? |
| Financial literacy of MSMEs | • Do women, youth, and disadvantaged people have access to financial education opportunities (such as academic courses, internships, and training courses)?<br>• Are there local, national, or international agencies that could provide local capacity-building opportunities to foster community business skills?<br>• Are there any mentoring and business incubation opportunities in the private sector for these groups?<br>• Are there local business advisory services for activities and enterprises in sustainable agriculture, agroforestry, forestry, other land uses, agro-processing, and natural infrastructure? |
| Access to finance technologies and services | • Do local financial institutions use digital technology to foster financial access for local communities?<br>• Do local communities use digital payments?<br>• What are the existing knowledge and skills (local, traditional, technical, scientific) in relation to the proposed economic activities?<br>• Are technical assistance mechanisms in place for MSMEs to transform production to more sustainable forms (e.g., fewer greenhouse gas emissions, no deforestation, conservation of water and soil resources, maintenance or enhancement of biodiversity)?<br>• What stakeholders in the landscape actively support technological innovation for agriculture and natural resource management and related value chain development?<br>• Do stakeholders in the landscape have access to the information (on climate, markets, supply of inputs and services, etc.) that is critical to their economic activities? |

**Table 2.** *Cont.*

| Framework Components | Guiding Questions |
|---|---|
| Facilities and mechanisms for inclusive integrated landscape finance | • Is there a clear and transparent mechanism between local landscape governance and finance facilities that ensures that financed projects and landscape objectives are aligned?<br>• Is there a solid pipeline of projects, whether bankable or not, that could create synergies at the landscape scale?<br>• Do financial institutions provide locally appropriate financial products and services?<br>• Do projects take a stakeholders' approach instead of a shareholders' approach?<br>• Are projects designed with a long-term perspective?<br>• What are the products, services, and ecosystem services generated by the projects?<br>• Are there markets for a diversity of products and services?<br>• How large, transparent, and volatile are the markets for the products and services?<br>• How do the projects create value in the landscape?<br>• For whom do they create value?<br>• Who would want to pay for or invest in the projects?<br>• Is it clear how projects will generate cash flows?<br>• Are benefit-sharing mechanisms in place that provide fair and transparent distribution of income from the proposed economic activities?<br>• What are the key financial and other risks to the project and finance actors? What risk mitigation strategies are available for projects and can they be implemented across projects? |

The questions in Table 2 would need to be validated according to their relevance across a range of landscape finance initiatives, but they can help proponents or evaluators of landscape finance reflect on the important elements of inclusive integrated landscape finance and how they relate to local conditions.

## 6. Conclusions

### 6.1. Challenges and Innovations in Addressing Them

While extensive literature exists on innovative finance for agriculture, on challenges to small-scale producers in access to finance, and on landscape initiatives, little has been written on how these issues can be addressed in a coordinated way to achieve inclusive finance for integrated landscape management. This gap reflects the emerging character, innovativeness, and complexity of the topic.

The review was useful in identifying eight common challenges to inclusiveness (Table 1): A product characteristic; B livelihood assets; C knowledge of agriculture and forestry sectors; D scale and costs of services; E transparency, trust, and governance; F production, climate, price, and policy risks; G information, communication, and roads; H other benefits, policies, and regulations. Different finance initiatives relevant to sustainable landscapes have addressed these challenges in different ways, depending on the local context: there is no one-size-fits-all solution. The innovations applied to address these challenges, however, could be grouped into four domains of enabling factors (Figure 3): inclusive landscape governance; strengthening the financial literacy of MSMEs; strengthening their access to technology and services; and the development of locally appropriate finance facilities and mechanisms. The review did not identify a single case that combined all these elements in a coordinated way. We suggest, therefore, that the framework for integrated landscape finance developed by the 1000 Landscapes Finance Solutions Team (Figure 1), while robust, needs to be adapted to more explicitly support financial inclusion by including the four domains (Figure 4). Assessing existing landscape finance initiatives

in relation to these four domains will help improve understandings of how to increase inclusiveness and create positive impacts in a range of contexts.

*6.2. Lessons Learned*

The review identified two main approaches to landscape finance. Top-down approaches come from outside the landscape, addressing international or national goals and/or value chain objectives to make local production sources more sustainable. Bottom-up approaches start from inside the landscape, working mainly with local savings and addressing the needs of local stakeholders. To reduce the dependence of landscape actors on a few outside financial flows, to increase their resilience to outside shocks—such as climate, market, or political changes—and to ensure that local stakeholders meet their own landscape objects, the two approaches need to be combined to achieve inclusive integrated landscape management.

Designing facilities and mechanisms that channel finance to inclusive integrated landscape management cannot, therefore, be left only to the financial sector. It requires an integrated approach, linking economic and financial expertise to technical and social knowledge and skills and to local practices and customs. In most cases, the collaboration of additional actors in the landscape is needed in order to achieve the desired results.

A key issue is whether the finance facilities adhere to the objectives of inclusive governance at the landscape level. This raises the question of which criteria and mechanisms to use to assess the performance and suitability (local and global) of the facility. Our proposed framework (Figure 4) can guide such assessments (Table 2), but we acknowledge that the framework is based on literature that offers information about a wide range of lessons learned but provides few examples of fully functional systems. More research is necessary to refine and test the framework and its application.

To move forward in designing finance facilities for inclusive integrated landscape management, it will be necessary both to identify and document existing cases and to implement and document new initiatives through interdisciplinary, transdisciplinary, and action research. Topics that require further research include:

- How should facilities for inclusive integrated landscape finance be structured, and what are successful strategies to make them relate to local inclusive governance mechanisms?
- What actors need to be involved to be able to implement inclusive integrated landscape finance, and how should they relate to each other?
- Who should pay for improvements in the enabling conditions that facilitate the involvement of local actors (e.g., strengthening capacities, building trust)? Will such payments ensure future integrated investments or do they favor only those involved in a particular agrocommodity value chain?
- How can inclusiveness, as well as scale, impact, and diversity of investments, be achieved in landscapes with different sizes and mixes of stakeholders?
- What combinations of financial products best align with local circumstances to achieve various types of non-financial benefits, including health, education, and ecosystem services?
- Do inclusive integrated finance facilities lead to more sustainable landscapes (e.g., low or negative greenhouse gas emissions, maintenance or enhancement of biodiversity and of locally essential ecosystem services, contributions to income and well-being)? How do we measure this in a transparent way that is understandable to all actors?

**Author Contributions:** Conceptualization, B.L., E.D.G. and L.G.P.; methodology, B.L., E.D.G. and L.G.P.; software, not applicable; validation, B.L., S.S., S.J.S., V.G., M.B. and A.M.; formal analysis, B.L., E.D.G. and L.G.P.; investigation, E.D.G. and L.G.P.; resources, B.L., M.B. and V.G.; data curation, B.L. and L.G.P.; writing—original draft preparation, B.L., E.D.G. and L.G.P.; writing—review and editing, B.L., A.M., V.G., S.S. and S.J.S.; visualization, B.L., E.D.G., L.G.P. and S.S.; supervision, B.L.; project administration, B.L.; funding acquisition, B.L., M.B. and V.G. All authors have read and agreed to the published version of the manuscript.

**Funding:** The study was coordinated by Tropenbos International within the framework of the Mobilising More for Climate and Working Landscape programs with the financial support of the Dutch Ministry of Foreign Affairs. It also received financial support from the CGIAR Research Program on Forests, Trees and Agroforestry (FTA) and from the NWO-WOTRO senior expert program of the Dutch Ministry of Foreign Affairs (grant 17953). Funding for EcoAgriculture staff inputs to the paper was generously provided by the Hitz Family Foundation.

**Institutional Review Board Statement:** Not applicable.

**Informed Consent Statement:** Informed consent was obtained from all people involved in the study.

**Data Availability Statement:** Not applicable. As it was a literature review, all material used was published prior to publication of this manuscript.

**Acknowledgments:** We gratefully acknowledge the support of Patricia Halliday in improving the language, Juanita Franco for the design of the diagrams, and the anonymous reviewers for their useful comments.

**Conflicts of Interest:** The authors declare no conflict of interest.

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
