# Peer review of "Access to Landscape Finance for Small-Scale Producers and Local Communities: A Literature Review"

_land, doi:10.3390/land11091444_

Round 1
Reviewer 1 Report
Review of: “Designing Landscape Finance to Include Small-scale Producers and Communities: Building a new conceptual framework”
This paper conducts a so-called integrative review of the literature on Landscape Finance, which the authors claim is an emerging area of research. The authors discuss the lit with the aim of detecting how access to Landscape Finance can be improved.
Comments
LANGUAGE
I rarely start a review by commenting on the English, but I feel it is absolutely in order for this manuscript. It is motivated to improve access to finance. But the irony is that the English in the manuscript is not very accessible.
In my view, the write-up would need to be simplified drastically, and be more focused. I can try to clarify this point by giving some examples. However, the point is general and applies throughout the paper.
Example 1 is the title of the manuscript. Usually the title of an article indicates what the paper does. The current title would suggest that the article builds a new conceptual framework for Landscape Finance. The current title suggest as well that this new conceptual framework is needed to include small-scale producers and communities in something. It is not clear in what exactly?… Based on the title, it seems small-scale producers and communities need to be included in either the design process, or in “the new conceptual framework”… both are not applicable). A better title would reflect that the manuscript is a lit review. Maybe a much better title would be “Designing Landscape Finance: A Literature Review”, Or “Landscape Finance: A Literature Review”, Or “Landscape Finance: An Integrative Literature Review”. (The last title is possibly already too complex as most readers will not know what an integrative lit review is.)
A second language point is that the write-up is too complex (throughout the article). There are too many acronyms, unhelpful labels for concepts, sentences that are too long, etc. Also, sentences often try to make more than one point, which is unhelpful.
Example 2: “We conducted an integrative literature review to identify the barriers that hinder, and factors that contribute to, inclusiveness of financial mechanisms for sustainable landscapes.”
This sentence tries to explain the aim of the paper (lit review); as well as a motivation for this aim (to identify…). Two points in one sentence! Even if you would like to make the two points in one sentence (which I think is a bad idea), then make it easier to understand! Maybe something like “We conduct a literature review to identify the factors that promote access to finance for projects that aim to create sustainable landscapes.”[1]
Example 3: Another early example “Agriculture, forestry and other land uses (AFOLU) is a key sector…” “Sector”? “Agriculture” and “Forestry” are sectors. But how are “other land uses” or “land-use” “sectors”? Such questionable labels make it very hard for me.
Actually the first sentence that made sense to me was this one: “In integrated landscape finance, any specific investment needs to be evaluated and designed in the context of the broader ‘landscape investment portfolio’” I take away from this sentence that externalities/spillovers are at the core of “integrated landscape finance”. Now, what would be “non-integrated landscape finance”? [Is there an important loss if we call everything simply “landscape finance”?
Finally, at critical stages precision is lacking. We read at the bottom of page 4:
“Overall, this led to 1496 potentially relevant publications. Of these we screened the abstract applying the same criteria as well as the following two.
i. Irrelevant topic assessed (for example finance for renewable energies, transport, cities, infrastructures, individuals).
ii. Geographic focus not relevant
How do you apply “geographic focus not relevant”? A little while later you say you focus on “applicable to tropical landscapes”. This confuses me. And are you really searching for articles with an “irrelevant topic”!!
Just earlier, in lines 155-156, you also say “Studies carried out in the Global North could be included in the ILR, if cases analyzed could be applied in tropical landscapes”. Does this mean that “Studies carried out in the Global North were always included in the ILR, if they were somehow “applicable” in tropical landscapes, or sometimes? What about studies “applicable” to the Global South but not to tropical landscapes? Or is the Global South identical to “tropical landscapes”, at least in your paper? Again, the point is, please be precise.
FOCUS OF THE PAPER
Frankly, after reading the paper, I am still not 100% sure what literature area the paper focuses on. I do find the start of page 5 helpful in this respect. Here you say “…final sample, addressing inclusive finance in a landscape context.” If this (i.e. “inclusive finance in a landscape context”) is indeed the actual focus then please let the reader know early on in the introduction (on page 1 or, at the latest page 2).
Maybe first define the term “landscape finance” after which you go on to make the point that your lit review focused on “Access to landscape finance.
SELECTION OF ARTICLES
Although you try to be very explicit how the articles got selected, I remain puzzled when it comes to the details. Lines 174-177 confused me, for example:
This led to a final selection of 141 publications selected for full text reading. Again, we applied the same criteria and added whether the document was electronically accessible through scopus, google scholar or google chrome. Eventually only 35 publications made it to the final sample, addressing inclusive finance in a landscape context.
How did the final sample of 141 articles still get reduced to 35 articles? Did you perhaps select articles based on “full-text article or working paper available in the public domain”? This would not be a satisfactory selection criterion according to me, seeing that most published articles are still not open access. [Or did you perhaps do something else? (I assume that “google chrome” means “the internet”?)
I would also say that a review of 35 articles is a little meager, even if this is an emerging field.
DEFINITION OF “SOLUTIONS” (“How to improve access to “landscape finance”?)
Table 2 introduces the “Conceptual Framework for designing landscape finance to include small-scale producers, communities, SMEs and FFPOs.”
For each of the suggested “solutions” suggested there I was left wondering how would we reach the listed desired outcomes. For example, Table 2 calls for multi-stakeholder partnerships at the landscape scale, as well as for “Transparent and well-established institutions and regulations.” But the question is always how are you going to create “Transparent and well-established institutions and regulations.”
In other words, isn’t the crux of the problem, one layer deeper, namely that we do not know for example how to create “Transparent and well-established institutions and regulations.” In other words, perhaps the authors identify the issues in Table 2, but not the solution to the issues. The difference between the issues and the solutions is important, but possibly the authors conflate the “issues” and “solutions to these issues”?
The authors need to realize that there is always a reason for the issues to arise. There is a reason for lack of coordination between stakeholders, for example, so saying that stakeholders should coordinate is wishful thinking. What would be needed is a change that would incentivize stakeholders to start coordinating with each other.
SCOPE OF THE “SOLUTIONS” (“objectives”, as would call them)
Even if we accepted that the solutions are really practical solutions, I am still also confused about the scope of the suggested “solutions”. Are these solutions really specific to “landscape finance”, hence “improving access to landscape finance”? Or would these “solutions”, when adequately implemented, also promote investments into other projects, for example unsustainable projects, non-landscape finance projects.
Perhaps this remarks reveals that I am yet to be convinced that “landscape finance” is really different from other forms of finance (at least when it comes to promotes these forms of finance). If the objectives in Table 2 were implemented, for example, it would presumably spur lending activity and therefore economic growth in the local landscape, more broadly speaking. It would be unclear if new lending would support sustainable, or rather both sustainable and unsustainable growth opportunities.
Let me also mention microfinance in this context. The article mentions microfinance a couple of times. These days, there is a sizable literature on microfinance. What can we learn about “landscape finance” from the body of knowledge on (i) promoting microfinance and (ii) optimally designing microfinance instruments?
[1] I hope that the authors realize that even “sustainable landscapes” need not be an obvious concept to readers. Sustainability can mean many different things, but the term has not been defined.

Reviewer 2 Report
this is a well written paper and a thorough review of literature and technique to determine what literature was pertinent.
the only item i saw was line 789 where the word "below" is used for table 2. I am in the habit of not using such a word when the document may scroll.
Author Response
Dear reviewer,
Thank you for your positive response. We have removed the word “below”.
Reviewer 3 Report
The paper proposes a framework for inclusive landscape finance for integrated landscape management. This topic is related to the journal's scope. I recommend making the following changes to the paper.
1. I suggest the authors provide an outline of the research paper’s structure for the readers.
2. Introduction section must widely be improved. Also, it is a mixed between Introduction and literature review. Likewise, the latter must be explicitly shown.
3. The objective must be explicitly indicated. Lines 116-131.
4. The authors can create some strong context and background for the readers. What has been done using the Perspectives and Challenges in inclusive finance field specifically until now? This should be included in the Introduction.
5. What is the difference between google scholar and google chrome? Line 176
6. How is social finance included in the framework? The framework is vast generalize.
7. Please elaborate in a suitable way the main differences with other frameworks proposes.
8. Is the proposed framework related to SDGs? How?
9. As part of your review of available literature, authors have to examine previous papers published in Land, cite those articles as part of your submission. Also, papers published at Sustainability.
10. Section 4.2 must be widened.
11. Further research must explicitly be indicated.
12. Please check references guidelines.
Round 2
Reviewer 1 Report
Goal of the paper (and structure of the introduction)
The goal of the paper has to be stated early on in the paper. As it stands, readers have to wait until line 227 to learn about the goal:
This paper reviews the factors that contributed to the inclusiveness of existing innovative financial structures and mechanisms targeted to sustainable and resilient landscapes. Drawing from…
As it is, everything before the start of 1.1 is a motivation for the study. Then 1.1 is sort of more information, which can be viewed of as an extended motivation. 1.2 is really clarifying how the question of the paper is answered, but the reader does not know the question itself until line 227. The introduction is consequently not strong, while it is lengthy.
I would like to request that the authors split the introduction into two sections, where the first is the actual introduction. This would communicate the message of the paper much better. Maybe it’s as simply as creating the second section (perhaps “The importance of integrated landscape finance”?) based on the bulk of 1.3 (maybe everything before line 227.
Key terms. In this revised version, the article does an acceptable job defining key terminology (it was well below par earlier). Yet, I am still going to ask for additional clarification on key terminology.
AFALU. I am still a little puzzled by the term “AFALU sector”. First of all, these seem multiple sectors, not just a single sector. So can we call it AFALU sectors, first of all?
Also, I hope the authors appreciate that readers do not generally have a firm grasp of the industry classification the authors have in mind and therefore may interpret the “AFALU sectors” as literally “All land uses sectors”. Here is why this would be a reasonable interpretation. AFALU stands for Agriculture, Forestry and Other Land Uses. So Other would be all but Agriculture and Forestry, logically making AFALU “All Land Uses”, right?
n I trust however that this is not about “All Land Uses” and wonder how the reader can be informed better regarding which “other land use sectors” we are talking about. Perhaps provide some example, or be more specific in your definition.
n Even if it is “All Land Uses” then still it would help to explain what other land uses there are, maybe by describing which sectors do not count as “land uses” sectors.
n Is there also a geographic focus? There is of course as is becoming clear in Section 3.1. Make sure to clarify what is the geographic scope in the introduction (and I suggest even the abstract)?
Landscape Finance and related. Landscape Finance (LF) is still not defined, until later in the introduction (in Section 1.3). Also, I am not sure whether there is a need to make thing more complex by having three terms, namely “Landscape Finance”, “Integrated Landscape Finance (ILF)”, and then Innovative Landscape Finance.
Readers need a short and sweet definition of LF or ILF upfront, i.e. very early on in the introduction. Key in the definition of ILF is that it incorporates the spin-offs/externalities across projects. Or, in other words, that the focus is on the social returns, rather than the private returns only. Or perhaps, more vaguely put, ILF considers a more holistic view on returns, rather than merely the private returns.
“Inclusiveness” and separating “credit rationing” and the “financing gap”. The authors define “Inclusiveness” as follows:
“Inclusiveness” here refers to facilitating access to finance for small-scale producers, 97 community-based enterprises, and for micro, small and medium enterprises (MSMEs, as 98 defined by the International Finance Corporation) [16] that have a direct link with local 99 agricultural or forest production. We use the term MSMEs to describe this target group.
In all settings projects get denied finance. After all financiers may deem projects non-viable (the private return is insufficient). Denying applications for funding happens in developed as well as less-developed countries, in personal and commercial loans, and in all sectors of the economy. However, and this is important, denying a credit application is not necessarily the same as a lack of access of finance. After all financial projects have to be viable and funding providers need to screen projects as such. The authors therefore are presumably taking about what is called “credit rationing” in the literature, i.e. denying credit to viable projects? Or is it something else that the authors are talking about, something that is focused on social returns, rather than private returns? Can the authors:
-- relate “Inclusiveness” to these concepts, “denying credit”, “credit rationing”, and “social returns” (or “externalities”); and
-- clarify if, and, if so, how they would empirically distinguish “lack of inclusiveness” and “denying credit to non-viable projects”?
Article selection
Section 3.1 relates closely to the selection of the articles piece (line 259-294). In fact, a quicker read almost suggests that the selection procedure for the articles has been described twice, namely on lines 259-294 and in Section 3.1. Please integrate these two pieces and this will clear things up for most readers.
The description also still somewhat unclear to me, especially the last reduction of the article selection (line 289):
Several scientific papers — whether open access or not — could not be accessed because the publication was not available online. Eventually, 35 publications made it to the final sample for the actual review. In addition to the criteria noted above, we obtained this final sample by adding one specific criterion: “Does it address the challenges of financing integrated landscape management?”
You’re introducing two more criteria here. About how many “scientific papers” (article fell through because they were not available online? And how many because they did not address “challenges of financing integrated landscape management”? I am not sure what to say here. 141 was already a smallish number of articles. This reduction to 35 papers is disappointing, obviously, as the 141 seemed to be already a focused set of papers. It almost appears to me that in this step a lot of arbitrariness happed, for example, that papers got removed if they did not see ILF as currently practices as a problem.
It is also not clear why you needed to read the 100+ articles in full to conclude that they they did not address “challenges of financing integrated landscape management”? Wouldn’t that take very little time to assess, namely a quick read of the introduction and perhaps the conclusion?
I would say that these final reduction steps need to anyway be incorporated into step (v). After all, it is not completely relevant if you read all the downloadable articles (among the 141) in full, or rather just the 35 papers you ended up selecting.
Finally, in your response to my comments on your first submission, you say:
“The main results (Table 1 and what is now Figure 3) are based on the initial scopus search, while the text of section 3 also presents the result of the “snowball” search guided by the specific issues resulting from the search in Scopus. Thus, the final number of reviewed articles goes well beyond 35 (which is in line with the character of an integrative literature review).”
Two comments:
First, I completely missed the description of the “result of the “snowball” search”. I suppose it is there, but there may be a reason I missed it. (Readers have just limited time!) Can you improve this description, please, and also incorporate it in Table 1, which, after all, deals with the selection of the papers.
Second, why do you need to make the point that your review is an integrative lit review? I realize I asked this earlier as well. In your revision you downplayed the “integrative” part, but why not eliminate it altogether? As I argued earlier your first submission was close to impossible to read. With the revision the readability has definitely improved, but it would be a mistake to think that this article is an easy read. Unless it is essential that the lit review is an “integrative lit review” (and you fail to show that it is), just speak in language that everybody gets, so “lit review”. This way people can focus on the essence of your paper, which deals with Landscape Finance (this essence and the concept of landscape finance are already hard enough to grasp)
Conclusion
I find the conclusion very hard to read. I think of the conclusion as a possible place where the key findings of the paper are described in plain language, for the current paper arguably language simple enough for “policy makers” (i.e. those described in line 101, namely: NGOs, companies, financial institutions, international organizations and government agencies) to understand the recommendations to make Landscape Finance more accessible. I do not believe the current version of the conclusion is very helpful in this respect.
Maybe very practically speaking, would you be able to focus the conclusion, so that it clearly addresses the questions you ask on lines 111-114?:
• What are the challenges to the inclusiveness of integrated landscape finance as a path to resilient landscapes?
• What innovations are being used to address those challenges?
• What can we learn from successful innovations and how can these lessons be applied in design and assessment of mechanisms for landscape finance?
Referee report on version 1
I observe that several points above were also made in my report on you manuscript version 1. I realize you provided a reply to the report, but it was a little short and not all the points were addressed adequately in the end. Can you please review my earlier report if you end up submitting a version 3 to the manuscript?
Smaller remarks.
The authors speaks of “medium enterprises”. The common term is “medium-sized enterprises”.
Line 141: “Such method can add value…” Method??
Line 207: “Whereas innovative finance structures are being developed to draw more money to the agriculture, forest and other land use sector, these often follow the more conventional approaches towards finance, focusing on financial return, risk and scale, and, more recently, transparent reporting of the impacts of the investments. Applying such financial structures for ILF may, therefore, not always solve the challenge of increasing access to the financial mechanisms for small-scale producers, communities, indigenous people and women and youth, since investments in their initiatives are often perceived to be less profitable, riskier and of too small a scale.”
I hope the authors realize that this defies logic? Theoretically, the improved conventional approaches can solve the issue. I think I understand what you are trying to say, and the point you are trying to make is a good one. However, it helps if the writing is based on logical statements; else the manuscript risks losing impact as it possibly appears to be too normative to readers.
Line 265: “Only studies published from 2010 on were considered”. You mean "from 2010 onwards"? How about “Only studies published between January 2010 and June 2021 were considered”
Table 1 and Sections 3.2.1-3.2.3. In Table 1 you distinguish between “Recipients of finance” and “Finance providers”. But then in Section 3.2.3 you suddenly introduce two more categories, namely Investors and producers.
n Do we need the Investors and producers at all? Why aren’t they in Table 1?
n What is the relationship between Investors and producers and the recipients and the providers? Investors are presumably a subset of the providers? And producers a subset of the recipients? Explain the relationship if we need the “new” categories.
Author Response
Thank you for your additional and helpful comments. Please see attached file for a detailed response to your comments

Reviewer 3 Report
The paper can be published in the present form.
Author Response
Thank you for your positive response!